# Prevalence of Chromosomally Located $bla_{CTX-M-55}$ in *Salmonella* Typhimurium ST34 Isolates Recovered from a Tertiary Hospital in Guangzhou, China

Shihan Zeng,[a,b] Zhenxu Zhuo,[a] Yulan Huang,[a] Jiajun Luo,[a] Yulian Feng,[a] Baiyan Gong,[a] Xiyi Huang,[d] Aiwu Wu,[b] Chao Zhuo,[c] Xiaoyan Li[a]

aDepartment of Clinical Laboratory, Fifth Affiliated Hospital, Southern Medical University, Guangzhou, China
bGuangzhou Key Laboratory for Clinical Rapid Diagnosis and Early Warning of Infectious Diseases, KingMed School of Laboratory Medicine, Guangzhou Medical University, Guangzhou, China
cState Key Laboratory of Respiratory Disease, First Affiliated Hospital, Guangzhou Medical University, Guangzhou, China
dDepartment of Clinical Laboratory, The Affiliated Shunde Hospital of Guangzhou Medical University (Lecong Hospital of Shunde District), Foshan, China

Shihan Zeng, Zhenxu Zhuo, and Yulan Huang contributed equally to this article. Author order was determined by the corresponding author after negotiation.

**ABSTRACT** Nontyphoidal *Salmonella* (NTS) is one of the most prevalent bacterial causes of gastrointestinal infections worldwide. Meanwhile, the detection rate of CTX-M-55 ESBL-positive has increased gradually in China. To identify the molecular epidemiological and genomic characteristics of $bla_{CTX-M-55}$-carrying nontyphoidal *Salmonella* (NTS) clinical isolates, a total of 105 NTS isolates were collected from a Chinese tertiary hospital. Antimicrobial susceptibility testing was performed to determine the resistance phenotype. Whole-genome sequencing and bioinformatics analysis were used to determine the antimicrobial resistance genes, serotypes, phylogenetic relationships, and the genetic environment of the $bla_{CTX-M-55}$ gene. The results showed that among the 22 ceftriaxone resistant isolates, the $bla_{CTX-M-55}$ was the most common $\beta$-Lactamase gene carried by 14 isolates, including serotypes *S.* Typhimurium (10/14), *S.* Muenster (2/14), *S.* Rissen (1/14), and *S.* Saintpaul (1/14). Phylogenetic analysis shows that 10 $bla_{CTX-M-55}$-positive *S.* Typhimurium ST34 isolates were divided into two clusters. The genetic relationship of isolates in each cluster was very close ($\leq$10 cgMLST loci). The $bla_{CTX-M-55}$ gene was located on the chromosome in 10 isolates, on IncI1 plasmid in three isolates, and IncHI2 plasmid in one isolate. In conclusion, the $bla_{CTX-M-55}$ gene, mainly located on the chromosome of *S.* Typhimurium ST34 isolates, was the main driving force associated with the resistance of NTS to cephalosporins. Therefore, close attention to the clonal dissemination of $bla_{CTX-M-55}$-carrying *S.* Typhimurium ST34 in clinical settings must be monitored carefully.

**IMPORTANCE** ESCs are the first choice for treating NTS infections. However, ESBLs and AmpC $\beta$-lactamases are the most typical cause for ESCs resistance. The CTX-M-55 ESBL-positive rate has gradually increased in the clinic in recent years. At present, the research about $bla_{CTX-M-55}$-positive *Salmonella* mainly focuses on the foodborne animals or the environment while less on clinical patients. Thus, this study was carried out for identifying molecular epidemiological and genomic characteristics of $bla_{CTX-M-55}$-carrying NTS clinical isolates. The results showed that the $bla_{CTX-M-55}$ gene, mainly located on the chromosome of *S.* Typhimurium ST34 isolates from Conghua District, was the main driving force associated with the resistance of NTS to cephalosporins. Therefore, our work highlights the importance of monitoring the clonal dissemination of $bla_{CTX-M-55}$-carrying *S.* Typhimurium ST34 in clinical settings.

**KEYWORDS** *S.* Typhimurium ST34, CTX-M-55, IncHI2, chromosome, cephalosporins

Address correspondence to Aiwu Wu, aiwwu66@163.com, Chao Zhuo, Chao_sheep@263.net, or Xiaoyan Li, xiaoyanli@gzhmu.edu.cn.

The authors declare no conflict of interest.

Nontyphoidal *Salmonella* (NTS) consists of more than 2600 serotypes and is one of the most prevalent bacterial causes of gastrointestinal diseases worldwide (1). In the Global Burden of Disease Study 2010 of the Institute for Health Metrics and Evaluation, enteric NTS was estimated to cause 4.8 million disability-adjusted life years and 81,300 deaths (2, 3). NTS serotypes are diverse in their host range and epidemiology while varying in their propensity to cause severe human diseases (4). The most widely reported serotypes associated with gastrointestinal diseases across China are *S.* Typhimurium and *S.* Enteritidis (5). In the United States, *S.* Enteritidis is the most common serotype associated with human foodborne diseases, followed by *S.* Typhimurium, *S.* Newport, *S.* Heidelberg, and *S.* Montevideo (6). Though most cases of human NTS diarrheal are self-limiting and usually do not require treatment. However, antibiotic treatment is necessary when invasive infection occurs, besides for children and elderly patients (7, 8). Ampicillin, trimethoprim-sulfamethoxazole, and chloramphenicol are the preferred first-line therapeutic antimicrobial agents recommended in the clinic (9). With the emergence and rapid development of resistance phenotype ACSSuT (defined as resistance to ampicillin, chloramphenicol, streptomycin, sulfamethoxazole, and tetracycline) in the early 1980s in *Salmonella*, fluoroquinolones (FQs) and extended spectrum cephalosporins (ESCs) have been used widely to treat NTS infection (4, 9–12).

ESCs are the first choice for treating NTS infections because FQs use contraindication in children (5). Therefore, the development of NTS isolates resistant to ESCs, such as ceftriaxone, represents a substantial public health concern (4). Generally, the productions of extended-spectrum $\beta$-lactamases (ESBLs) and AmpC $\beta$-lactamases are the most typical causes for ESCs resistance (13). Since the first report of *Escherichia coli* CTX-M ESBLs in 1989, the CTX-M family has spread worldwide and became the most prevalent ESBLs (14, 15). The CTX-M-15 and CTX-M-14 are the most common variants detected in clinical pathogens all over the world, followed by CTX-M-2, CTX-M-3 and CTX-M-1 (16). Previous data suggest that ESCs resistance genes in *Salmonella* were mainly *bla*CMY-2 and *bla*CTX-M-15 (4, 13, 17–19). A study conducted in Thailand (20) found that the *bla*CTX-M-55 was the vital ESCs resistance mechanism for *S.* Cholerae in NTS from 2012 to 2016. CTX-M-55 is a variant of CTX-M-15, which possesses similar hydrolysis activities to ESCs. However, enhanced activity of CTX-M-55 was found for ceftazidime (15). CTX-M-55 was first found in *E. coli* in Thailand in 2005 (15) and first detected in *Salmonella* in China in 2011 (21). Recently, the detection rate of CTX-M-55 identified in ESBLs *E. coli* has increased gradually worldwide, especially in China, irrespective of the clinic, livestock industry, and environment (22–29). In several studies (16, 23, 30), the detection rate of CTX-M-55 in *E. coli* has exceeded the rate of CTX-M-15, becoming the second prevalent ESBLs in China. The incidence rate of CTX-M-55-producing bacteria has increased significantly in both animals and humans (22, 31). Available data suggest that the *bla*CTX-M-55 gene is located on the epidemic plasmids, such as IncF, IncI1, IncHI2, and IncA/C carried by *Salmonella* (25, 31–33). To date, many researchers have focused on CTX-M-55-producing *E. coli*, whereas only a few studies have focused on the CTX-M-55-producing *Salmonella*, especially isolates isolated from clinical patients.

Clinically, the detection rate of *Salmonella* increases sharply in June and July each year in our hospital. Notably, some NTS isolates were resistant to ESCs. Therefore, the objective of this study was to analyze the prevalence and antimicrobial resistance mechanisms of cephalosporins in NTS isolates obtained from a tertiary hospital in China.

## RESULTS

**Bacterial strains.** From May 21, 2020, to February 22, 2021, 105 NTS isolates were isolated from patients in a Chinese tertiary hospital. The number of NTS isolates collected in different months is shown in Fig. 1. The highest detection rate of NTS was in July, accounting for 30.48% (32/105). The detection rate from December to February was relatively low, accounting for only 8.57% (9/105). Among the 105 patients, there were 56 males and 49 females, with a male to female ratio of 1.14:1. The isolation

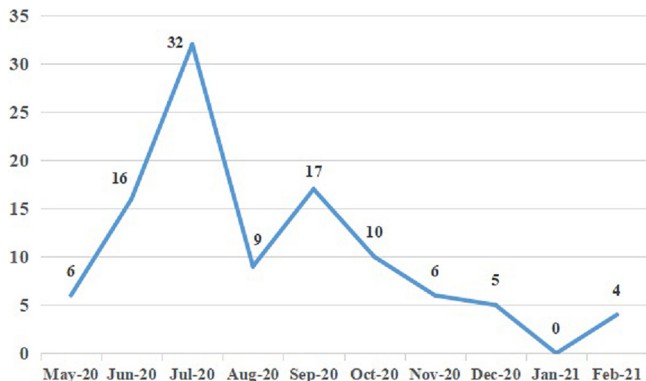

**FIG 1** The trend of the incidences of NTS isolates.

sources include stool (95.24%, 100/105), blood (2.86%, 3/105), purulent secretion (0.95%, 1/105) and urine (0.95%, 1/105). The age of patients ranged from 14 days to 83 years, of which 61.90% were ≤ 1 year old, 15.24% were 2–5 years old, 5.71% were 6–17 years old, 11.43% were 18–59 years old, and 5.71% were ≥ 60 years old. A total of 14 $bla_{CTX-M-55}$-positive isolates (Table S2) were detected in 105 NTS. Among these, the patient's age ranged from 2 months to 83 years, of which 57.14% (8/14) were ≤1 year old. All $bla_{CTX-M-55}$-positive isolates were derived from the stool sample. Of note, 50% of $bla_{CTX-M-55}$-positive strains were isolated in July (Table S2).

**Serotype.** A total of 13 serotypes were identified among the 105 NTS isolates (Fig. 2). Among these isolates, *S.* Typhimurium was the most prevalent serotype (66.67%, 70/105), followed by *S.* Enteritidis (8.57%, 9/105), *S.* London (6.67%, 7/105), *S.* Rissen (4.76%, 5/105), and *S.* Stanley (3.81%, 4/105). Collectively, these serotypes accounted for 90.48% (95/105) of NTS isolates in the present study. Additionally, among 70 isolates of *S.* Typhimurium, 56 were ST34, and 14 were ST19.The serotypes of $bla_{CTX-M-55}$-positive isolates consisted of *S.* Typhimurium ST34 (10/14), *S.* Muenster ST321 (2/14), *S.* Rissen ST469 (1/14), and *S.* Saintpaul ST27 (1/14).

**Phylogenetic analysis of $bla_{CTX-M-55}$-carrying isolates.** In the ceftriaxone resistant isolates (*n* = 22), 14 carried the $bla_{CTX-M-55}$ gene (Table S2). Among them, $\beta$-lactamase genes $bla_{TEM-1}$ (5/14) and $bla_{LAP-2}$ (1/14) were also detected. Meanwhile, the $\beta$-lactamase

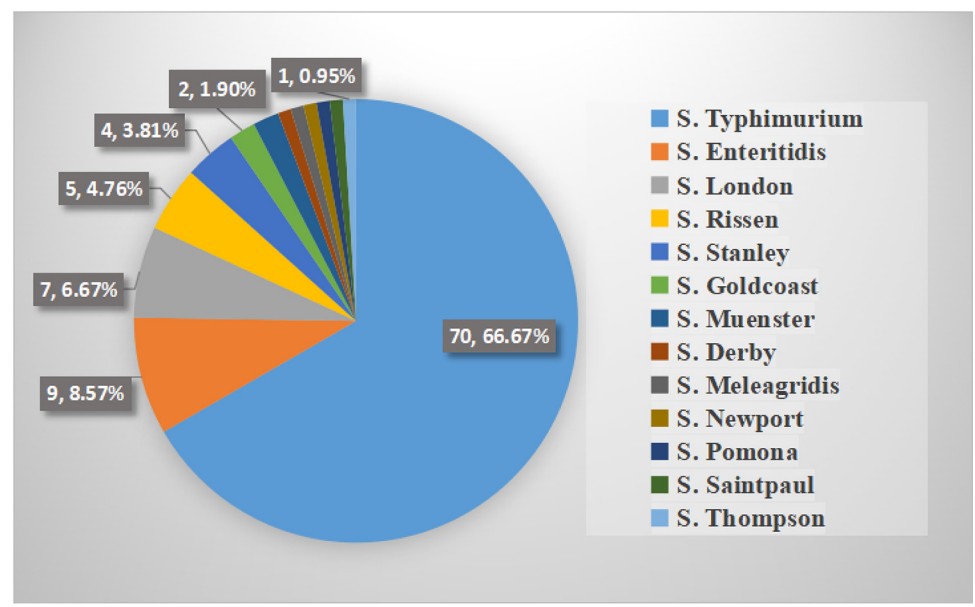

**FIG 2** The serotype distribution of the NTS isolates.

coding genes, including $bla_{CTX-M-65}$, $bla_{CTX-M-123}$, $bla_{TEM-1}$, $bla_{CMY-2}$, $bla_{OXA-1}$, and $bla_{OXA-10}$, were detected in the 8 ceftriaxone resistant isolates (Table S2). The CTX-M type was the most abundant $\beta$-Lactamases, accounting for 86.36% (19/22). On the other hand, the detection rate of $bla_{CTX-M-55}$ gene was 63.64% (14/22), which was the most common ESBLs.

The minimum spanning tree (MST) based on the cgMLST allele profile of $bla_{CTX-M-55}$-positive *S.* Typhimurium ST34 isolates showed that it could be divided into two clusters. The genetic relationship of isolates in each cluster was closely related ($\leq$10 alleles) (Fig. S1). Interestingly, these strains were isolated from the stool sample of patients living in different streets. Four strains of cluster 1 were isolated in July, and the other four strains of cluster 1 were from June, August, September, and November, respectively (Table S2). The two strains of cluster 2 were isolated from patients of different ages and different departments of hospitalization. The results indicated that the *S.* Typhimurium ST34 strain carrying the $bla_{CTX-M-55}$ gene has a clonal epidemic trend in this district, likely to be part of an uncovered outbreak epidemic event. Moreover, cgMLST analysis revealed only five allele differences between the two *S.* Muenster strains (Fig. S2). The whole-genome sequence of 22 *S.* Typhimurium ST34, 19 S. Rissen ST469, 2 S. Muenster ST321, and 1 S. Saintpaul ST27 isolates recovered from China were downloaded from EnteroBase (https://enterobase.warwick.ac.uk/species/index/senterica, retrieved in February 2022). The strain metadata was listed in Table S3. The phylogenetic analysis revealed that our $bla_{CTX-M-55}$-positive *S.* Typhimurium ST34 isolates differed by the 137~220 SNPs with other Chinese *S.* Typhimurium ST34 isolates deposited in the public database (Fig. 3B). In this study, the number of SNP differences in the 56 strains of *S.* Typhimurium ST34 ranged from 2 to 160 (Fig. S3). In addition, the resistance genes carried by the $bla_{CTX-M-55}$-positive *S.* Typhimurium ST34 were not enriched by other Chinese *S.* Typhimurium ST34.

**Antibiotic susceptibility and resistance genes detection.** Among all the isolates, 86.7% were resistant to AMP, followed by CHL (40%) and SXT (39%) (Table S4). Interestingly, the resistance rates of NTS isolate to CRO, CAZ and FEP were 21%, 18.1%, and 13.3%, respectively. Meanwhile, the sensitivity of NTS isolates for CIP reduced, in which the resistance rate was only 3.8%, with the intermediary rate of 56.2%. All isolates were susceptible to IPM and LVX, while most isolates (94.3%) presented a high susceptible rate to AZM. Genomic data revealed that (i) all AMP resistant strains carried at least one ESBLs or AmpC gene (including $bla_{TEM}$, $bla_{CTX-M}$, $bla_{OXA}$, $bla_{LAP}$, and $bla_{CMY}$); (ii) 92.7% (38/41) of CHL resistant strains carried at least one gene responsible for phenicol resistance (including *floR*, *cmlA*, and *cat*); (iii) all SXT resistant strains carried at least one gene that causing co-trimoxazole resistance (including *dfrA* and *sul*); (iv) 77.8% of CIP insensitive strains carried at least one of the genes of *aac(6')-Ib-cr*, *oqxA*, *oqxB*, *qepA8*, *qnrB*, *qnrS1*, and *qnrS2*; (v) and all AZM resistant strains carried *mph*(A) gene which mediates macrolide resistance.

All of the $bla_{CTX-M-55}$-positive isolates were resistant to the CXM, CRO, CAZ, and AMP but unsusceptible to FEP, as the resistance rate of FEP was 78.6% (Table S5). The sensitivity of $bla_{CTX-M-55}$-positive isolate for CIP was also decreased, with an intermediate rate of 78.6%. Additionally, the resistance rates of $bla_{CTX-M-55}$-positive isolates for CHL and SXT were 21.4%, while the resistance rate to AZM was only 7.1%. However, all $bla_{CTX-M-55}$-positive isolates were sensitive to IPM and LVX. The 10 $bla_{CTX-M-55}$-positive isolates were *S.* Typhimurium ST34, out of which the resistance phenotype of 7 isolates was "AMP+CXM+CAZ+CRO+FEP", and the remaining 3 isolates were "AMP+CXM+CAZ+CRO" (Table 1). The resistance phenotype of the two *S.* Muenster isolates carrying $bla_{CTX-M-55}$ gene was "AMP+CXM+CAZ+CRO+FEP+CHL". Thus, $bla_{CTX-M-55}$ gene mediates the high level of FEP resistance. Furthermore, compared with *S.* Typhimurium ST34, the clinical treatment of the other three serotypes carrying $bla_{CTX-M-55}$ gene was more challenging. Meanwhile, the number of resistance genes carried by $bla_{CTX-M-55}$-positive isolates with serotypes *S.* Muenster, *S.* Rissen, and *S.* Saintpaul were significantly higher than those in *S.* Typhimurium ST34 (Table 1). For example, *cmlA1* and *floR* genes were not detected in *S.* Typhimurium ST34 but the other three serotypes. Of note, 11 (78.57%) of $bla_{CTX-M-55}$-positive isolates also carried the *qnrS* gene

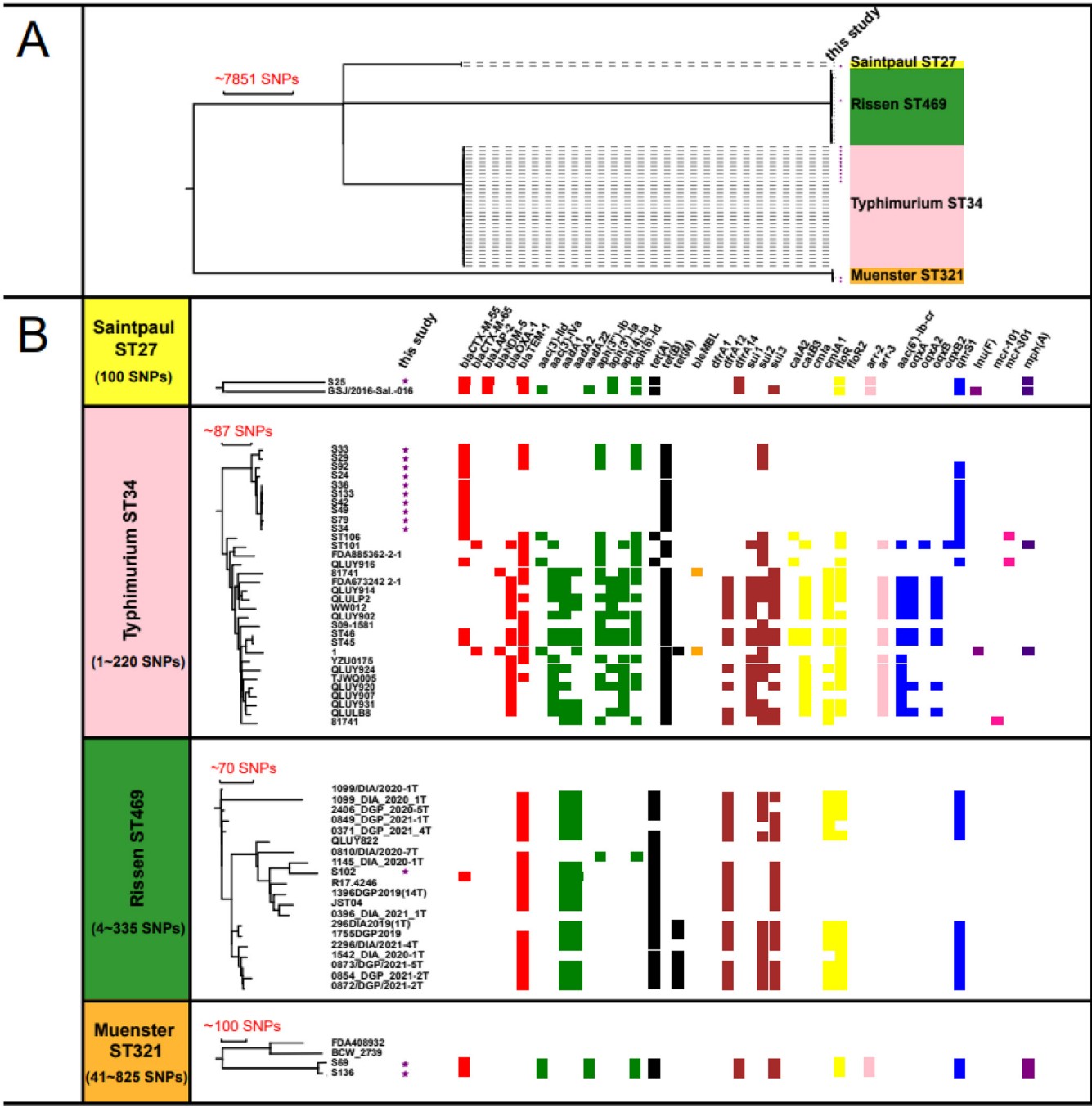

**FIG 3** Phylogeny and the distribution of antimicrobial resistance genes of *bla*<sub>CTX-M-55</sub>-positive isolates from the present study and NTS isolates from China contained from the EnteroBase database. (A) Systematic analysis of *bla*<sub>CTX-M-55</sub>-positive isolates in this study and NTS isolates from China contained in the EnteroBase database. (B) SNP difference range and resistance gene distribution among strains of various serotypes. The strains from this study were marked with purple stars. Different colored boxes represent the resistance genes for the different types of antibiotics, while the blank boxes indicate the resistance genes which were not detected.

mediating FQs resistance, of which all isolates had lesser sensitivity to CIP (Table 1). Importantly, in this study, the two *S.* Muenster strains only carried one plasmid of ColRNAI, as most of the resistance genes were carried on their chromosome.

**The location and genetic environment of *bla*<sub>CTX-M-55</sub> gene.** For most *bla*<sub>CTX-M-55</sub>-positive isolates (10/14), the *bla*<sub>CTX-M-55</sub> gene was located on the chromosome. In addition, the *bla*<sub>CTX-M-55</sub> gene of three isolates was situated on the IncI1 plasmid and the gene of one strain was located on the IncHI2 plasmid (Table 1). Therefore, the

**TABLE 1** The genomic characteristics of bla_CTX-M-55-carrying isolates

| Isolates | Serotype | ST | Location of bla_CTX-M-55 | Plasmids | Resistance phenotype | Resistance genes[a] |
|---|---|---|---|---|---|---|
| S25[b] | Saintpaul | 27 | IncHI2 | IncHI2 | AMP+CXM+CAZ+CRO+FEP+CHL+AZM | bla_CTX-M-55, qnrS, bla_LAP-2, bla_TEM-1, aph(3')-Ia, aph(6)-Id, arr-2, dfrA14, floR, mph(A), tet(A) |
| S102 | Rissen | 469 | IncI1 | IncI1, Col440II, ColRNAI | AMP+CXM+CAZ+CRO+FEP+SXT | bla_CTX-M-55, bla_TEM-1, aadA1, aadA2, dfrA12, sul3, tet(A) |
| S29[b] | Typhimurium | 34 | IncI1 | IncI1, IncQ1, Col440II, Col8282, ColRNAI | AMP+CXM+CAZ+CRO+FEP | bla_CTX-M-55, bla_TEM-1, aph(3')-Ib, aph(6)-Id, sul2, tet(B) |
| S33 | Typhimurium | 34 | IncI1 | IncI1, IncQ1, Col156, Col440II, Col8282, ColRNAI | AMP+CXM+CAZ+CRO+FEP | bla_CTX-M-55, bla_TEM-1, aph(3'')-Ib, aph(6)-Id, sul2, tet(B) |
| S24 | Typhimurium | 34 | Chromosome | ColRNAI | AMP+CXM+CAZ+CRO+FEP | bla_CTX-M-55, qnrS, tet(B) |
| S34[b] | Typhimurium | 34 | Chromosome | \[c] | AMP+CXM+CAZ+CRO+FEP | bla_CTX-M-55, qnrS, tet(B) |
| S36 | Typhimurium | 34 | Chromosome | \ | AMP+CXM+CAZ+CRO+FEP | bla_CTX-M-55, qnrS, tet(B) |
| S42 | Typhimurium | 34 | Chromosome | Col440I | AMP+CXM+CAZ+CRO+FEP | bla_CTX-M-55, qnrS, tet(B) |
| S92 | Typhimurium | 34 | Chromosome | IncQ1 | AMP+CXM+CAZ+CRO+FEP | bla_CTX-M-55, qnrS, bla_TEM-1, aph(3'')-Ib, aph(6)-Id, sul2, tet(B) |
| S49 | Typhimurium | 34 | Chromosome | \ | AMP+CXM+CAZ+CRO | bla_CTX-M-55, qnrS, tet(B) |
| S79 | Typhimurium | 34 | Chromosome | \ | AMP+CXM+CAZ+CRO | bla_CTX-M-55, qnrS, tet(B) |
| S133 | Typhimurium | 34 | Chromosome | \ | AMP+CXM+CAZ+CRO | bla_CTX-M-55, qnrS, tet(B) |
| S69 | Muenster | 321 | Chromosome | ColRNAI | AMP+CXM+CAZ+CRO+FEP+CHL+SXT | bla_CTX-M-55, qnrS, aac(3)-IId, aadA22, aph(6)-Id, arr-2, dfrA14, floR, lnu(F), sul3, tet(A) |
| S136 | Muenster | 321 | Chromosome | ColRNAI | AMP+CXM+CAZ+CRO+FEP+CHL+SXT | bla_CTX-M-55, qnrS, aac(3)-IId, aadA22, aph(6)-Id, arr-2, dfrA14, floR, lnu(F), sul3, tet(A) |

[a]Underlined genes were located on chromosomes (only sequenced strains on the Nanopore platform were analyzed).
[b]The isolate was selected and sequenced for long read sequencing on Nanopore platform.
[c]\ indicates no plasmid.

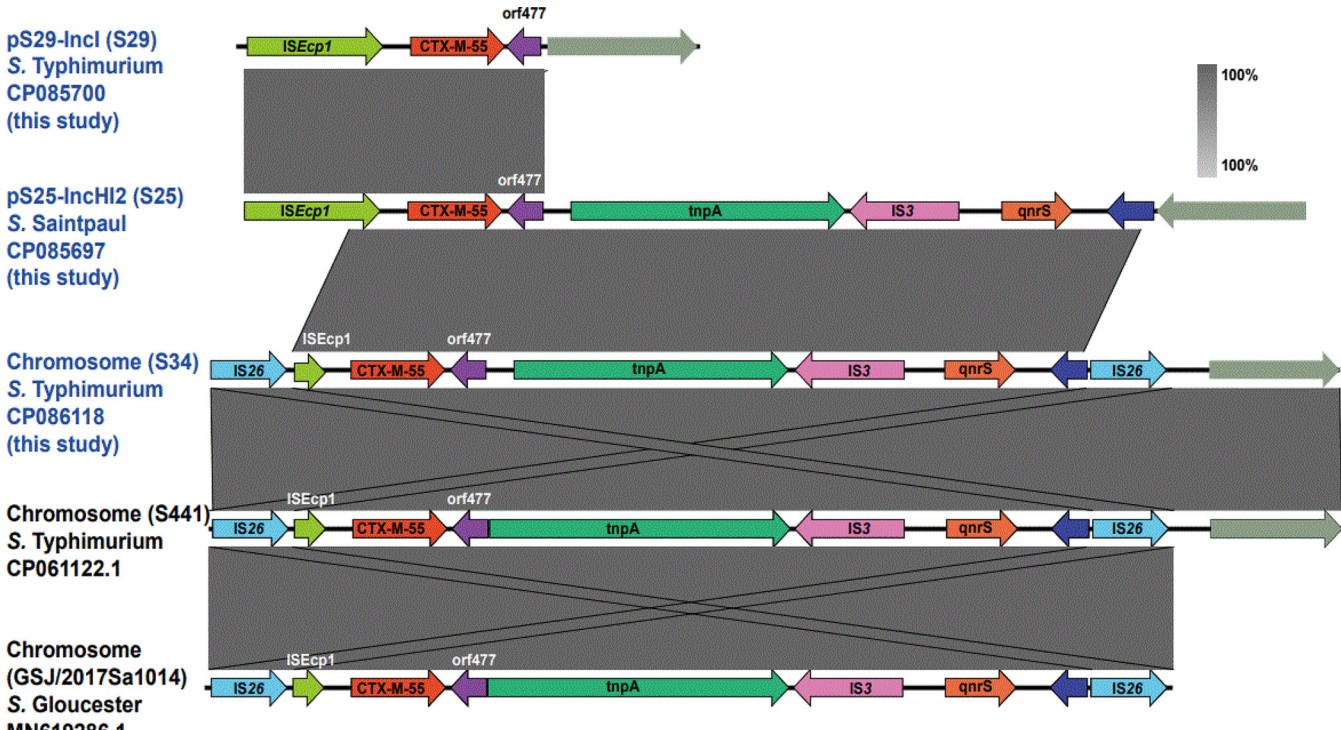

**FIG 4** The genetic environments of *bla*<sub>CTX-M-55</sub>-positive isolates. Different colored arrows indicate different open reading frames (ORFs), while the direction of the arrow indicates its transcription direction. Light gray shadows represent homologous areas. The isolates from the current study were identified in blue. IS*Ecp1* located on the chromosome of isolate S34 was disrupted by IS*26*, while IS*Ecp1* in pS29-IncI1 and pS25-IncHI2 plasmid were complete.

*bla*<sub>CTX-M-55</sub> gene located on the IncI1 plasmid carried by the two *S.* Typhimurium ST34 isolates (S29 and S33) could be successfully conjugated into the recipient. Consistent with the original donor strains, the MIC of the two transconjugants for CRO, CAZ, and FEP were ≥ 64mg/L, ≥64 mg/L, and 16 mg/L, respectively (Table S5).

The genetic context of *bla*<sub>CTX-M-55</sub> gene was different on the IncI1 plasmid, IncHI2 plasmid, and the chromosome (Fig. 4). The mobile elements located upstream of *bla*<sub>CTX-M-55</sub> included IS*Ecp1* (complete or incomplete) and IS*26*, while the downstream of *bla*<sub>CTX-M-55</sub> gene was a hypothetical protein named ORF477. The genetic structure of the *bla*<sub>CTX-M-55</sub> gene on pS29-IncI1 (CP085700) in isolate S29 was "IS*Ecp1*-*bla*<sub>CTX-M-55</sub>-orf477." Additionally, the *qnrS* gene was located downstream of *bla*<sub>CTX-M-55</sub> gene on the IncHI2 plasmid and the chromosome of the *bla*<sub>CTX-M-55</sub>-positive NTS isolates. The genetic environment of the IncHI2 is similar to that of the chromosome, the main difference being that IS*26* is located at both ends of the chromosome, and IS*Ecp1* is truncated by an IS*26*. IS*Ecp1* upstream of the *bla*<sub>CTX-M-55</sub> gene on IncHI2 was complete. (Fig. 4). The search in the NCBI database revealed that the chromosomes of *S.* Typhimurium (CP061122.1) strain isolated from a patient in Zhejiang, China, and *S.* Gloucester (MN619286.1) strain isolated from roast duck products in Guangdong, China, also contained this genetic structure "IS*26*-ΔIS*Ecp1*-*bla*<sub>CTX-M-55</sub>-orf477-tnpA-IS*3*-*qnrS*-IS*26*." Importantly, the IS*26* composite transposon seems to play an essential role in the emergence and transmission of *bla*<sub>CTX-M-55</sub> between chromosome and IncHI2 plasmid.

Interestingly, the sequence similarity of IncI1 plasmid pS29-IncI1and pEC32-IncI1 (CP085621) was up to 99.94% by BLAST (Fig. 5), carrying only one resistance gene *bla*<sub>CTX-M-55</sub>. In our previous study, pEC32-IncI1 was the vital plasmid carrying *bla*<sub>CTX-M-55</sub> gene found in *E. coli* isolate EC32. When searching the NCBI GenBank database, we found the sequence identity between the several IncI1 plasmids carrying *bla*<sub>CTX-M-55</sub> gene isolated from *E. coli* (pWP3-S18-ESBL-09, AP022037.1), *Klebsiella pneumonia* (pKP4823_3, CP082793.1), and *S.* Typhimurium (pST53-2, CP050747.1) was higher than 99% (Fig. 5). In our previous study, the pEC71-IncHI2 (CP085623) plasmid carrying the *bla*<sub>CTX-M-55</sub> gene was found in the *E. coli*, which was mainly had a similar structure of

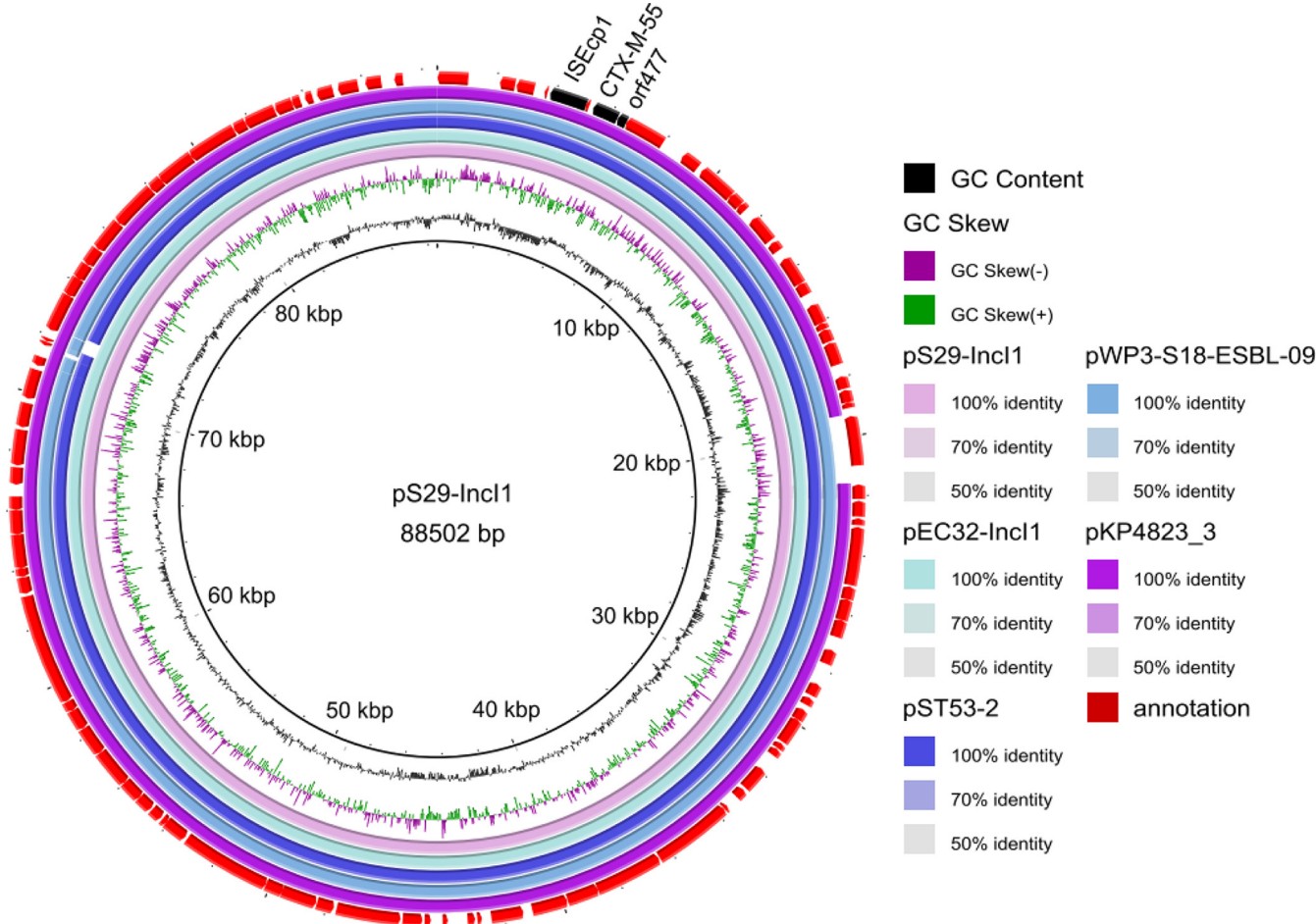

**FIG 5** The circle diagram of IncI1 plasmid carrying *bla*<sub>CTX-M-55</sub> gene in *Salmonella* isolate S29 and *E. coli* isolate EC32. The inner circle is GC content, the second circle is GC skew, the third circle to the seventh circle are different IncI1 plasmid, and the outermost circle represents the annotation. The reference sequence used was pS29-IncI1. The sequence identity between the fiveIncI1 plasmids was higher than 99%. The position of the "IS*Ecp1-bla*<sub>CTX-M-55</sub>-orf477" genetic structure is marked in black.

IncHI2 plasmid pS25-IncHI2 (CP085697) isolated from *S.* Saintpaul isolate S25 in this study by BLAST (Fig. 6). Interestingly, the fragment carrying the *bla*<sub>CTX-M-55</sub> gene (about 19000 bp) between the two IncHI2 plasmids pS25-IncHI2 and pEC71-IncHI2 had a reverse orientation, which also contains *qnrS* and *dfrA14* genes. In addition, the *bla*<sub>CTX-M-55</sub> gene flanking sequence contained in the two IncHI2 plasmids was the same. However, compared with pS25-IncHI2, pEC71-IncHI2 lacked one IS26 transposon in the reverse fragment (Fig. 6).

## DISCUSSION

NTS infection is an important public health concern. Ours and previous data suggest that NTS infection frequently occurs in summer, and the risk groups are infants, the elderly, and males (34–36). It may be due to the warm weather and concentrated rainfall in summer, suitable for NTS reproduction, and easily polluting food (35, 36). Consumption of contaminated food or water is the main route of NTS transmission (37).

*S.* Typhimurium has a variety of hosts and is one of the most common serotypes for foodborne illnesses in humans and animals (38). Many studies have revealed *S.* Typhimurium as the most common serotype in China in the pediatric and adult population (5, 39). In this study, 13 serotypes were identified in the 105 NTS isolates. Among them, *S.* Typhimurium was the most common serotype, accounting for almost 66.67% of all the isolates, much higher than the second-largest serotype *S.* Enteritidis (8.57%). More than 60 STs were found in *S.* Typhimurium worldwide, out of which ST34

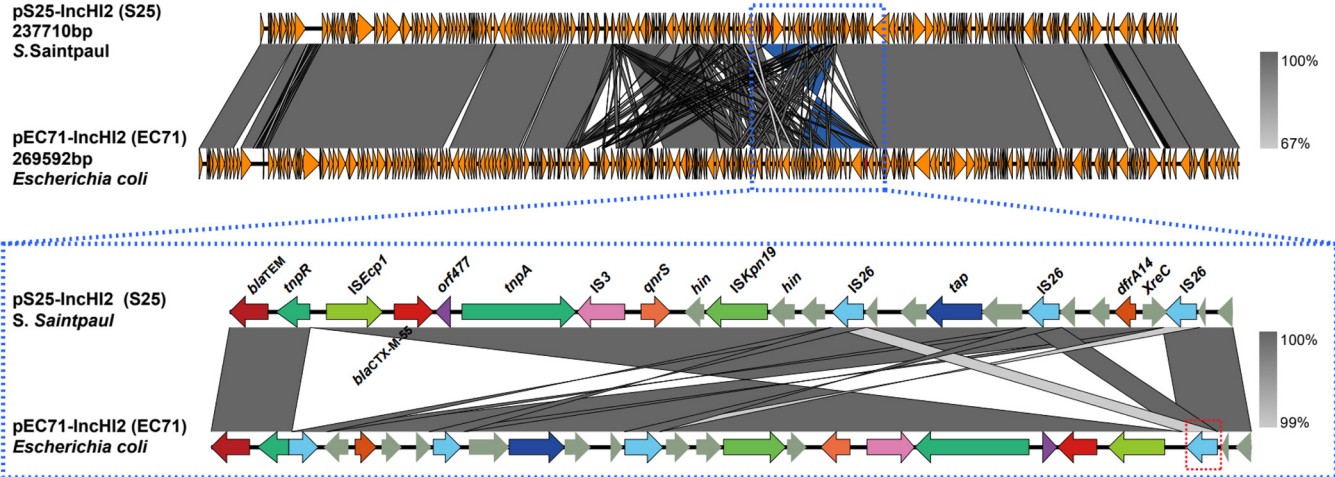

**FIG 6** The gene fragment containing *bla*<sub>CTX-M-55</sub> on the IncHI2 plasmid. The light gray shadow indicates the homologous region, and the blue box shows the gene fragment containing the *bla*<sub>CTX-M-55</sub> gene. The *bla*<sub>CTX-M-55</sub> gene fragment contained in plasmid pS25-IncHI2 of *Salmonella* isolate S25 and plasmid pEC71-IncHI2 of *E. coli* isolate EC71 are similar IS*26* added upstream of IS*Ecp1* of pEC71-IncHI2 (indicated by red dotted box).

(57%) and ST19 (28.4%) were the most common STs in Asia alone (40). In this study, only two STs were found in *S*. Typhimurium, of which 80% were ST34, and 20% were ST19. This study indicates that *S*. Typhimurium was the leading serotype in the district, where ST34 was the main clone.

After the severe resistance from NTS, FQs and ESCs were extensively used to treat the NTS infection (9–11). While *Salmonella* infection tends to occur more in children, the FQs are contraindicated in this age group. Therefore, ESCs remain the primary treatment of choice for NTS in children. Notably, a high level of resistance to ESCs has been reported, usually related to the plasmid-mediated transmission of AmpC or ESBLs genes (11). Among the ceftriaxone-resistant isolates collected in this study, the *bla*<sub>CTX-M-55</sub> gene had the highest detection rate. The *Salmonella* isolated from foodborne infection in the animals in Guangzhou, China (24). The *bla*<sub>CTX-M-55</sub> gene was also the most common ESBLs gene. In addition, the *bla*<sub>CTX-M-55</sub> gene was still the major ESBLs in the analysis of *S*. Typhimurium from patients with diarrhea in Shanghai, China (41). Likewise, similar reports have been reported from Cambodia and Thailand (20, 25).

Different NTS serotypes are evident for host predilection (42). Most of the *bla*<sub>CTX-M-55</sub>-positive isolates in this study were *S*. Typhimurium ST34 (10/14). However, it was previously reported (24) that the most common serotype of the *bla*<sub>CTX-M-55</sub> gene of *Salmonella* in Chinese foodborne animals was *S*. Indian (7/12). The *S*. Typhimurium ST34 has attracted international attention due to its rapid transmission and multiple drug resistance (MDR). Moreover, previous studies showed that although *S*. Typhimurium ST34 showed rapid expansion in China, it was sensitive to ESCs (43). However, the *bla*<sub>CTX-M-55</sub> gene in *S*. Typhimurium caused a high resistance level for the ESCs in this study. As an effective choice for the treatment of NTS infection, the current resistance situation of ESCs is not optimistic. Surprisingly, the phylogenetic analysis of *bla*<sub>CTX-M-55</sub>-positive *S*. Typhimurium isolates indicated the *S*. Typhimurium ST34 strain with *bla*<sub>CTX-M-55</sub> gene on the chromosome has a clonal epidemic trend in this district. Other serotypes carrying the *bla*<sub>CTX-M-55</sub> gene were also detected in the study, although in a relatively smaller number. *S*. Muenster is often seen in cattle and has rare in humans (44, 45). In most cases, the *S*. Muenster was often associated with fatality rates (46, 47). Our study showed that the two *S*. Muenster strains were closely related and carried multiple resistance genes on the chromosomes. This observation may highlight the associated significant challenges to clinical treatment. However, because the numbers of *S*. Muenster were relatively small in this study, it is difficult to interpret whether it is an epidemic clone. The MDR of *S*. Saintpaul S25 in this study might be mediated by the plasmid pS25-IncHI2, which carries many antimicrobial resistance genes. Furthermore, the *bla*<sub>LAP-2</sub> gene, with the narrow

hydrolysis spectrum against $\beta$-lactams inhibited by clavulanic acid (48), was also detected in isolate S25. Meanwhile, isolates S25 carried the *mph*(A) gene, which might have a high resistance to azithromycin (49). However, azithromycin gained the approval of the US Food and Drug Administration (FDA) in 2011 as an additional drug for treating *Salmonella* infection (50). The emergence of the *mph*(A) gene in NTS also brings corresponding challenges to public health. *S.* Rissen is a common serotype distributed in the swineherds, pork, and chicken products, most of which are MDR strains, having a strong ability to infect humans (51, 52). With the emergence of multidrug resistant NTS, public health faces significant challenges. Therefore, NTS isolates carrying the *bla*~CTX-M-55~ gene should be critically monitored.

The conjugative plasmid is an important factor for the emergence and transmission of resistance genes. A study in Cambodia (25) showed that the transmission of *S.* Enteritidis of the *bla*~CTX-M-55~ gene was mediated by IncA/C2 (14/26) and IncHI2 (10/26) plasmids. However, the study on *bla*~CTX-M-55~ on *Salmonella* for foodborne infection of animals in Guangzhou (24) showed that 1 of 11 isolates was located on the IncHI2 plasmid. Among the *bla*~CTX-M-55~-positive isolates in this study, in addition to chromosome and IncHI2 plasmid, the *bla*~CTX-M-55~ gene of three isolates was located on IncI1 plasmid. Our previous reports have shown that the IncI1 plasmid was a predominant plasmid carrying *bla*~CTX-M-55~ gene in *E. coli* in this district, while most of them can be transferred by conjugation experiment. We also found that the sequence of IncI1 plasmid carrying *bla*~CTX-M-55~ gene in *Salmonella* and *E. coli* was almost similar (99.94%), whereas they both carried only one resistance gene*bla*~CTX-M-55~. IncI1 plasmid was one of the most common plasmids carrying the *bla*~CTX-M-55~ gene in *E. coli* (26). In this study, the IncI1 plasmid located in the two *S.* Typhimurium ST34 strains could be transferred through a conjugation experiment, indicating that the IncI1 plasmid might play a vital role in transmitting the *bla*~CTX-M-55~ gene in Enterobacteriaceae. A study (53) on the surveillance of antibiotic resistance plasmids in *Salmonella* isolates showed that the IncHI2 was the main plasmid lineage, helping spread antibiotic resistance in *Salmonella*. The plasmid pS25-IncHI2 carried multiple resistance genes in this study, including *bla*~CTX-M-55~, *bla*~TEM~, *qnrS*, *dfrA14*, and *tet*(A). Interestingly, pS25-IncHI2 could not be transferred through the conjugation experiment; however its role in the MDR of clinical *Salmonella* isolates should not be underestimated.

Generally, the genes responsible for cephalosporin resistance of clinical isolates are mainly associated with the plasmids (11). However, the frequency of the *bla*~CTX-M-55~ gene of NTS isolates on chromosomes was much higher than that of plasmids in this study. We found the high frequency (10/14) of the *bla*~CTX-M-55~ gene on the NTS chromosome. Among them, 8 strains with the *bla*~CTX-M-55~ gene located on the chromosome were *S.* Typhimurium ST34, and all belong to cluster 1 by cgMLST analysis (differential allele ≤10). Of note, most *bla*~CTX-M-55~-carrying isolates also harbored the *qnrS* gene, which can mediate low-level FQs resistance. Although PMQR genes only confer low-level FQs resistance, their presence (especially *qnrS*) provides strains with selective advantages under FQs exposure. It can also accelerate the development of chromosome-mediated drug resistance (24, 54). Though the high frequency of the *bla*~CTX-M-55~ gene on the chromosomes has been reported previously in Guangzhou (11/12), the isolation source was from foodborne animals, and the most frequent serotype was *S.* Indian (7/11), followed by *S.* Typhimurium (2/10) (24). Unfortunately, the above study lacks the analysis of the *bla*~CTX-M-55~ genetic environment on the chromosome. Furthermore, the strains of cluster 1 came from different patients, coming from different subdistrict, and the dates of isolates collection were also different (Table S2). The phylogenetic tree also showed that the *bla*~CTX-M-55~-positive *S.* Typhimurium ST34 isolated from the Conghua district had large SNP different from other Chinese *S.* Typhimurium ST34 (Fig. 3). These results suggest that the *S.* Typhimurium ST34 strain carrying *bla*~CTX-M-55~ gene has a clonal epidemic trend in this district or was likely to be an outbreak epidemic event. Considering the relatively few strains in this study, we will continue to collect clinical isolates for follow-up research. Some studies speculate

that the cross-species transmission of the $bla_{CTX-M-55}$ gene from a plasmid in *E. coli* to the *Salmonella* chromosome was conducive to the transmission and stability of the $bla_{CTX-M-55}$ gene in *Salmonella* (24, 55). Therefore, close observation is required for the clonal dissemination of $bla_{CTX-M-55}$-chromosomally located *S*. Typhimurium ST34 in clinical settings.

By analyzing the genetic environment of $bla_{CTX-M-55}$ on IncHI2 and chromosome, we found that the genetic environment of $bla_{CTX-M-55}$ gene on IncHI2 and chromosome is almost the same (about 6500 bp); the difference is that there is an IS*26* composite transposon in the genetic environment of chromosome "IS*26*-ΔIS*Ecp1*-$bla_{CTX-M-55}$-orf477-tnpA-IS*3*-qnrS-IS*26*." It is worth mentioning that the gene environment of 10 strains of $bla_{CTX-M-55}$ gene on the chromosome was consistent, including *S*. Muenster isolates. Interestingly, the above composite transposons were also detected on the chromosomes of *S*. Gloucester strain (32) isolated from Guangdong roast duck products in China and *S*. Typhimurium strain of a patient in Zhejiang, China. Furthermore, it is reported (56–59) that IS*26* plays a significant role in the movement, expression and transmission of antibiotic resistance genes (ARGs) in Gram-negative bacteria (especially Enterobacteriaceae). This strain can mediate the transfer of resistance genes through various transposition modes. The most common transposition mode is the intermolecular transposition of the composite transposon (16). The composite transposon consists of two copies of IS*26*. Therefore, we speculated that the transposon event mediated by IS*26* composite transposon might be an important event for transferring $bla_{CTX-M-55}$ gene from IncHI2 plasmid to chromosome and its occurrence on the chromosomes of different serotypes of *Salmonella*. Combining the two IS*26* elements can form a composite transposon, which may mediate the reversal of the $bla_{CTX-M-55}$ gene fragment on pEC71-IncHI2and pS25-IncHI2. A study (56) reported various ARGs and multiple IS*26* copy regions on the composite transposon bounded by IS*26*. Overall, IS*26* composite transposon-mediated transposition plays a vital role in transferring the $bla_{CTX-M-55}$ gene of NTS.

**Conclusion.** *S*. Typhimurium ST34 is the primary pathogen of NTS infection in a Chinese tertiary hospital. The $bla_{CTX-M-55}$ gene was the most common $\beta$-Lactamase gene in the NTS and was mainly located on the chromosome. Notably, the *S*. Typhimurium ST34 strain with $bla_{CTX-M-55}$ gene on the chromosome has a clonal epidemic trend in the studied period. Furthermore, IncI1 plasmid, IncHI2 plasmid, and IS*26* composite transposon-mediated transposition also play an essential role in transferring the $bla_{CTX-M-55}$ gene in *Salmonella*. Meanwhile, the $bla_{CTX-M-55}$ and qnrS gene cotransfer warrants additional attention because it may accelerate the development and spread of isolates coresistant to cephalosporins and fluoroquinolones. Most importantly, close attention needs to be paid to monitor the clonal dissemination of $bla_{CTX-M-55}$-carrying *S*. Typhimurium in clinical settings.

## MATERIALS AND METHODS

**Bacterial strains.** From May 21, 2020, to February 22, 2021, 105 NTS nonrepetitive strains were isolated from different patients in the Fifth Affiliated Hospital of Southern Medical University, China. The isolates were identified by Vitek-2 COMPACT automatic microbial identification system (bioMérieux, Marcy-l'Étoile, France). *E. coli* ATCC 25922 was used as the quality control strain.

**Antimicrobial susceptibility testing.** MICs for ESBLs-resistant phenotype were determined using the Vitek-2 Automated Susceptibility System (60). The MICs of NTS isolates for amoxicillin-clavulanic acid (AMC), sulfamethoxazole (SXT), cefuroxime (CXM), ceftriaxone (CRO), ceftazidime (CAZ),cefepime (FEP), cefoxitin (FOX), imipenem (IPM), and levofloxacin (LVX) were determined by broth microdilution. The susceptibility of NTS isolates for azithromycin (AZM), ampicillin (AMP), ciprofloxacin (CIP), and chloramphenicol (CHL) was determined by the Kirby Bauer disk diffusion method. The interpretation of the results was made based on the Clinical and Laboratory Standards Institute (CLSI/NCCLSM100-S30) (61).

**Whole-genome sequencing and bioinformatics analysis.** The genomic DNA of 105 NTS isolates was extracted by the bacterial genomic DNA extraction kit (Tiangen, Beijing, China) and then sequenced using Illumina NovaSeq 6000 platform (Novogene, Tianjin, China). The quality of the raw sequence reads was checked by the interactive program FastQC (62), assembled by SPAdes 3.13.1 (63), and annotated by Prokka 1.14.5 (64). The serotypes of NTS were analyzed using SISTR 1.1.1 (65) and the sequence type (ST) was analyzed by multilocus sequence typing (MLST 2.18.0) (66). The antimicrobial resistance genes and plasmids replicon type were determined based on the CGE server (67). CGE's ResFinder 4.1 and PlasmidFinder 2.1 web tools with default threshold settings for minimum % identity 90% and minimum % coverage 60% were used for analyses. The $bla_{CTX-M-55}$-positive NTS were selected for subsequent

analysis. The core genome multilocus sequence typing (cgMLST) of 10 $bla_{CTX-M-55}$-positive *S.* Typhimurium isolates was performed using Ridom SeqSphere+ 4.1.9 (68). Three representative $bla_{CTX-M-55}$-positive *S.* Typhimurium isolates were selected and sequenced on the Oxford Nanopore platform (Novogene, Tianjin, China), which $bla_{CTX-M-55}$ gene was located on the chromosome, IncI1 plasmid, and IncHI2 plasmid, respectively (Details of selection is shown in the Table S1). The hybrid genome assembly using both long and short reads was performed by Unicycler 0.4.8 (69). A phylogenetic tree was constructed using Parsnp (70) and was edited with MEGA X (71). Finally, the location and genetic environment of the $bla_{CTX-M-55}$ gene in $bla_{CTX-M-55}$-positive NTS were analyzed and illustrated with BRIG (72). The genetic context of the $bla_{CTX-M-55}$ gene was drawn by Easyfig (73).

**Plasmid conjugation experiments.** The transferability of the plasmid carrying the $bla_{CTX-M-55}$ gene was determined by the conjugation experiments with the rifampin-resistant *E.coli* C600 as the recipient strain. Transconjugants were selected on the Luria-Bertani agar plates containing rifampin (100 $\mu$g/mL) and CRO (4 $\mu$g/mL). PCR (Primers as follows: CTX-M-55-F is CAGCGCTTTTGCCGTCTAAG, and CTX-M-55-R is GGCCCATGGTTAAAATCACTGC) sequencing was used to verify the transconjugants containing the $bla_{CTX-M-55}$ gene. In addition, an antimicrobial susceptibility test was performed to confirm the antimicrobial resistance characteristics of these transconjugants.

**Nucleotide sequence accession number.** This Whole Genome Shotgun project has been deposited at DDBJ/ENA/GenBank under the accession ID: JAJGYQ000000000-JAJGZD000000000. The complete nucleotide sequences of plasmids pS25-IncHI2, pS29-IncI1, pEC32-IncI1, and pEC71-IncHI2 have been uploaded to the NCBI GenBank database under accession no. CP085697, CP085700, CP085621, and CP085623, respectively. The complete genomic sequence of isolate S34 that $bla_{CTX-M-55}$ gene located on the chromosome has been uploaded to NCBI GenBank database with accession no. CP086118.

## SUPPLEMENTAL MATERIAL

Supplemental material is available online only.
**SUPPLEMENTAL FILE 1**, XLSX file, 4.8 MB.
**SUPPLEMENTAL FILE 2**, PDF file, 0.2 MB.

## ACKNOWLEDGMENTS

We thank Zhi Ruan (Sir Run Shaw Hospital, Zhejiang University School of Medicine, Hangzhou, China) for his suggestions and revisions to this study, and Yan Jiang (Sir Run Shaw Hospital, Zhejiang University School of Medicine, Hangzhou, China) for his technical assistance in the genomic data analysis.

We declare that the research was conducted without any commercial or financial relationships that could be construed as a potential conflict of interest.

This project was funded by the National Natural Science Foundation for the Youth of China (81902104), the International Cooperation and Exchange Program of the National Natural Science Foundation of China (81861138056), and Guangdong Basic and Applied Basic Research Foundation (2021A1515220153).

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
