## [Reviewer comments · Microbiology Spectrum]

Microbiology Spectrum

Prevalence of Chromosomally Located *bla*_{CTX-M-55} gene in *Salmonella* Typhimurium ST34 clonal strains from Conghua District of Guangzhou, China

Shihan Zeng, Zhenxu Zhuo, Yulan Huang, Jiajun Luo, Yulian Feng, Baiyan Gong, Ai-Wu Wu, Chao Zhuo, and Xiaoyan Li

Corresponding Author(s): Xiaoyan Li, Department of Clinical Laboratory, Fifth Affiliated Hospital, Southern Medical University

Review Timeline:

Submission Date:	December 29, 2021
Editorial Decision:	February 1, 2022
Revision Received:	March 5, 2022
Accepted:	April 1, 2022

Editor: Sandeep Tamber

Reviewer(s): The reviewers have opted to remain anonymous.

Transaction Report:

DOI: <https://doi.org/10.1128/spectrum.02771-21>

February 1, 2022

Dr. Xiaoyan Li
Department of Clinical Laboratory, Fifth Affiliated Hospital, Southern Medical University
Guangzhou
China

Re: Spectrum02771-21 (**Prevalence of Chromosomally Located *bla*_{CTX-M-55} gene in *Salmonella* Typhimurium ST34 clonal strains from Guangzhou, China**)

Dear Dr. Xiaoyan Li:

Link Not Available

Sincerely,

Sandeep Tamber

Journals Department
Editor comments:

The focus, clarity, and presentation of the manuscript needs to be improved. Please expand on the background information provided in the Introduction and ensure the methods are described fully so they can be reproduced by others. The link below provides information on language editing services for your consideration. If these items in addition to the points below are properly addressed, I will consider the publication of this manuscript.

Reviewer comments:

Reviewer #1 (Comments for the Author):

To identify the epidemiology and the genetic environment of the rising pathogen CTX-M-55 *Salmonella* in China, authors collected 105 *Salmonella* strains that include 70 *S. Typhimurium*. Fourteen *Salmonella* were carrying *bla*_{CTX-M-55} gene 2 *S. Munster*, 1 *S. Rissen*, 1 *S. Saintpaul* and 10 *S. Typhimurium*. However, the late 10 strains were phylogenetically divided into 2 close clusters, pointy towards only 2 clones or strains instead of 10 strains. Authors suggested that these clusters should be considered as not detected outbreaks. Considering that, all analysis is based on 2 outbreak strains, all prevalence and epidemiological analysis do not reflect the real epidemiology. In a published study (2020), not cited in this publication, authors investigated the prevalence of *bla*_{CTX-M-55} in 4724 strains in China, concluding to the same finding as this study regarding the location, real prevalence and ST typing.

Authors, should consider publishing this work with another angle, as the number of strains does not reflect the title of the study. The text should be revised with regard to orthograph, examples:

Line 79: space missing between clinic and (6)

Line 112: E. coli should be in italic

Line 130: space missing between 29 and "and"

Line 132: space missing between server and (31)

Line 133: 2 spaces between for and subsequent

Reviewer #2 (Comments for the Author):

This manuscript investigated the prevalence of blaCTX-M-55 gene in Non-typhoidal Salmonella from a tertiary hospital between May 2020 and February 2021, and clarified the blaCTX-M-55 gene mainly located on the chromosome of S. Typhimurium ST34, highlighting the importance of monitoring the clonal dissemination of blaCTX-M-55-carrying S. Typhimurium ST34 in clinical settings.

In generally, the outcomes have clinical implications, but sample size is limited and further investigation is needed. The discussion should be more condensed and focused.

Major concerns:

(1) The authors claimed that blaCTX-M-55 is located on chromosomes or plasmids in 14 strains. Did they do Pulsed-field gel electrophoresis with S1 nuclease and Southern blotting experiments? How else would they select three representative strains for Nanopore sequencing? Please provide the detail information and data.

(2) Line 271-273: This point here is not appropriate. How could you get the point? There have no IS26 in plasmid IncHI2.

(3) The Figures in this manuscript are too simple and crude to contain much useful information, and are poorly made.

Minor comments:

(1) In line 32,33, 36, 37, there should be Spaces between words

(2) In line 128, the FastQC reference may not be quite correct, please double-check it.

(3) In lines 131-132, what are the parameters in the process of determining resistance genes and plasmid replication subtypes through the CGE server, for similarity and coverage, etc.? These should be stated.

(4) In line 683, 684, 686 and 687. ISECP1, E.coli, Salmonella as well as resistance genes should be italicized. Such errors also appeared elsewhere in the manuscript, please check.

Reviewer #3 (Comments for the Author):

This paper characterized 105 clinical Salmonella strains, the results should be more concentrate. For example, how many patients are the strains from? What's the meaning of the result of months and detection rate of male and female patients? This paper can be more important by just describe and discuss most important findings. Other problems are listed below.

1.Capitalize the first letter of the title?

2.Spaces are missing in many places

3.Line 112, the name of strain should be in Italic.

4.The results should be re-organized. It is better to have a separate result for antibiotic resistance genes in Line 188-195.

5.Phylogenetic analysis should contain more strains from China.

6.Many grammer problems.

7.Figure should be more clear.

Staff Comments:

Preparing Revision Guidelines

- Point-by-point responses to the issues raised by the reviewers in a file named "Response to Reviewers," NOT IN YOUR COVER LETTER.
- Upload a compare copy of the manuscript (without figures) as a "Marked-Up Manuscript" file.

- Each figure must be uploaded as a separate file, and any multipanel figures must be assembled into one file.
- Manuscript: A .DOC version of the revised manuscript
- Figures: Editable, high-resolution, individual figure files are required at revision, TIFF or EPS files are preferred

Please return the manuscript within 60 days; if you cannot complete the modification within this time period, please contact me. If you do not wish to modify the manuscript and prefer to submit it to another journal, please notify me of your decision immediately so that the manuscript may be formally withdrawn from consideration by Microbiology Spectrum.

This manuscript investigated the prevalence of *bla*_{CTX-M-55} gene in Non-typhoidal *Salmonella* from a tertiary hospital between May 2020 and February 2021, and clarified the *bla*_{CTX-M-55} gene mainly located on the chromosome of *S. Typhimurium* ST34, highlighting the importance of monitoring the clonal dissemination of *bla*_{CTX-M-55}-carrying *S. Typhimurium* ST34 in clinical settings.

In generally, the outcomes have clinical implications, but sample size is limited and further investigation is needed. The discussion should be more condensed and focused.

Major concerns:

(1) The authors claimed that *bla*_{CTX-M-55} is located on chromosomes or plasmids in 14 strains. Did they do Pulsed-field gel electrophoresis with S1 nuclease and Southern blotting experiments? How else would they select three representative strains for Nanopore sequencing? Please provide the detail information and data.

(2) Line 271-273: This point here is not appropriate. How could you get the point? There have no IS26 in plasmid IncHI2.

(3) The Figures in this manuscript are too simple and crude to contain much useful information, and are poorly made.

Minor comments:

(1) In line 32,33, 36, 37, there should be Spaces between words

(2) In line 128, the FastQC reference may not be quite correct, please double-check it.

(3) In lines 131-132, what are the parameters in the process of determining resistance

genes and plasmid replication subtypes through the CGE server, for similarity and coverage, etc.? These should be stated.

(4) In line 683, 684, 686 and 687. *ISECPI*, *E.coli*, *Salmonella* as well as resistance genes should be italicized. Such errors also appeared elsewhere in the manuscript, please check.

Spectrum02771-21

Response to Reviewers

Thanks all the reviewers for these precious comments and suggestions. According to reviewer's comments, we answered the questions one by one and showed the revision as follows:

1. Editor comments:

The focus, clarity, and presentation of the manuscript needs to be improved. Please expand on the background information provided in the Introduction and ensure the methods are described fully so they can be reproduced by others. The link below provides information on language editing services for your consideration. If these items in addition to the points below are properly addressed, I will consider the publication of this manuscript.

Authors' response:

Thank you for your time for the review of this article. We ensure that the method has been described fully and the background information in the introduction has been appropriately added (Added in the introduction and highlighted in yellow). In addition, the uploaded manuscript has used the language editing service as you recommended.

2. Reviewer #1 (Comments for the Author):

2.1 Reviewer#1 comments:

To identify the epidemiology and the genetic environment of the rising pathogen CTX-M-55 Salmonella in China, authors collected 105 Salmonella strains that include 70 S. Typhimurium. Fourteen Salmonella were carrying blaCTX-M-55 gene 2 S. Munster, 1 S. Rissen, 1 S. Saintpaul and 10 S. Typhimurium. However, the late 10 strains were phylogenetically divided into 2 close clusters, pointing towards only 2 clones or strains instead of 10 strains. Authors suggested that these clusters should be considered as not detected outbreaks. Considering that, all analysis is based on 2 outbreak strains, all prevalence and epidemiological analysis do not reflect the real epidemiology.

In a published study (2020), not cited in this publication, authors investigated the

prevalence of blaCTX-M-55 in 4724 strains in China, concluding to the same finding as this study regarding the location, real prevalence and ST typing.

Authors' response:

Thank you for the suggestion. According to your suggestion, we added the content of "Considering the relatively few of strains in this study, we will continue to collect clinical isolates for follow-up research." in lines 420-421 and highlighted in yellow. In addition, we have carefully read the literature you recommended (Global clonal spread of *mcr-3*-carrying MDR ST34 *Salmonella enterica* serotype Typhimurium and monophasic 1,4,[5],12:i:- variants from clinical isolates), and found that its research object is *mcr-3*-positive strains. There is no doubt that that is an excellent article, which is worth learning. And we cite it in line 107 and highlighted in yellow.

2.2 Reviewer#1 comments:

Authors, should consider publishing this work with another angle, as the number of strains does not reflect the title of the study.

Authors' response:

Thank you for the suggestion. According to the review's advice, we changed the original Title "Prevalence of Chromosomally Located *bla*_{CTX-M-55} gene in *Salmonella* Typhimurium ST34 clonal strains from Guangzhou, China" to "Prevalence of Chromosomally Located *bla*_{CTX-M-55} gene in *Salmonella* Typhimurium ST34 clonal strains from Conghua District of Guangzhou, China".

2.3 Reviewer#1 comments:

The text should be revised with regard to orthograph, examples:

Line 79: space missing between clinic and (6)

Line 112: *E. coli* should be in italic

Line 130: space missing between 29 and "and"

Line 132: space missing between server and (31)

Line 133: 2 spaces between for and subsequent

Authors' response:

Thank you for the suggestion. We carefully checked to ensure that the above

errors have been corrected.

3. Reviewer #2 (Comments for the Author):

3.1 Reviewer#2 comments:

The authors claimed that blaCTX-M-55 is located on chromosomes or plasmids in 14 strains. Did they do Pulsed-field gel electrophoresis with S1 nuclease and Southern blotting experiments? How else would they select three representative strains for Nanopore sequencing? Please provide the detail information and data.

Authors' response:

Thank you for the suggestion. According to your advice, we have shown the selection details on the Supplementary Table S1 (as shown in the figure below), and it has been added in line 148-149 and highlighted in yellow.

Among all the blaCTX-M-55-positive isolates, the *S. Saintpaul* S25 carries a resistant plasmid IncHI2, which means that blaCTX-M-55 may exist on this plasmid; The blaCTX-M-55 gene and IncI1 plasmid replicator of *S. Typhimurium* S29 and S33 are located on the same contigs, that is, the blaCTX-M-55 gene of both are located on IncI1. Moreover, both are Cluster 2; Six strains of *S. Typhimurium* were without resistant plasmid replicates (S24, S34, S36, S42, S49, S79, and S133), belonging to Cluster 1; Even if *S. Typhimurium* S92 carries IncQ1, the cgMLST results show that it also belongs to Cluster 1; Long-read sequencing of *S. Rissen* and *S. Muenster* was not performed due to limited funding. Taken together, we selected S25, S29 and S34 with blaCTX-M-55 gene localized on IncHI2, IncI1 and chromosomes, respectively, as representative strains.

Supplementary Table S1: Details of representative isolates selected for third-generation sequencing						
Isolates	Serotype	Contigs-CTX-M-55 ¹	bp ²	Resistant plasmid ³	Contigs-rep ⁴	cgMLST cluster
S24	Typhimurium	39	5048	/	/	Cluster 1
S25	Saintpaul	28	11033	IncHI2	19	No analysis
S29	Typhimurium	19	82174	IncI1	19	Cluster 2
S33	Typhimurium	19	82174	IncI1	19	Cluster 2
S34	Typhimurium	40	3715	/	/	Cluster 1
S36	Typhimurium	40	3715	/	/	Cluster 1
S42	Typhimurium	40	3715	/	/	Cluster 1
S49	Typhimurium	41	3714	/	/	Cluster 1
S69	Muenster	24	10553	/	/	Cluster 3
S79	Typhimurium	39	3715	/	/	Cluster 1
S92	Typhimurium	40	3713	IncQ1	38	Cluster 1
S102	Rissen	25	79847	IncI1	25	No analysis
S133	Typhimurium	40	3715	/	/	Cluster 1
S136	Muenster	24	10131	/	/	Cluster 3

1: Number of contigs carrying blaCTX-M-55 gene in FASTA file after assembling of strain second generation data; 2: Sequence length of contig carrying blaCTX-M-55 gene; 3: Resistant plasmid replicator detected in strain; 4: Number of contigs carrying resistant plasmid replicator in FASTA file after assembling of strain second generation data.

3.2 Reviewer#2 comments:

Line 271-273: This point here is not appropriate. How could you get the point? There have no IS26 in plasmid IncHI2.

Authors' response:

Thank you for the suggestion. In our manuscript, we want to express IS26 at both ends of the genetic framework on the chromosome, not at the ends of the IncHI2 plasmid. Sorry for misleading you, so we reorganized content and changed it to "The genetic environment of the IncHI2 is similar to that of the chromosome, the main difference being that IS26 is located at both ends of the chromosome, and ISEcp1 is truncated by an IS26. ISEcp1 upstream of the *bla*_{CTX-M-55} gene on IncHI2 was complete (Fig. 4)", which in line 285-288 now and highlighted in yellow.

3.3 Reviewer#2 comments:

The Figures in this manuscript are too simple and crude to contain much useful information, and are poorly made.

Authors' response:

Thank you for the suggestion. According to your suggestion, we have made corresponding changes in line 219-231, 705-712, and figure 3, which highlighted in yellow.

3.4 Reviewer#2 comments:

In line 32,33, 36, 37, there should be Spaces between words;

In line 683, 684, 686 and 687. ISECP1, E.coli, Salmonella as well as resistance genes should be italicized. Such errors also appeared elsewhere in the manuscript, please check.

Authors' response:

Thank you for the suggestion. We carefully checked to ensure that the above errors have been corrected.

3.5 Reviewer#2 comments:

In line 128, the FastQC reference may not be quite correct, please double-check it.

Authors' response:

Thank you for the suggestion. It has been changed in line 136 and highlighted in yellow.

3.6 Reviewer#2 comments:

In lines 131-132, what are the parameters in the process of determining resistance genes and plasmid replication subtypes through the CGE server, for similarity and coverage, etc.? These should be stated.

Authors' response:

Thank you for the suggestion. It has been changed in line 140-142 and highlighted in yellow. The sentence added is "CGE's ResFinder 4.1 and PlasmidFinder 2.1 web tools with default settings, threshold for minimum % identity 90% and minimum % coverage 60%, were used for analyses."

4. Reviewer #3 (Comments for the Author):

4.1 Reviewer#3 comments:

This paper characterized 105 clinical Salmonella strains, the results should be more concentrate. For example, how many patients are the strains from? What's the meaning of the result of months and detection rate of male and female patients? This paper can be more important by just describe and discuss most important findings.

Authors' response:

Thank you for the suggestion. A total of 105 NTS non-repetitive strains were isolated from different patients in the Fifth Affiliated Hospital of Southern Medical University, which in line 117-119 and highlighted in yellow. We have added relevant contents in the discussion section; it has been added in line 313-318 and highlighted in yellow.

4.2 Reviewer#3 comments:

- 1.Capitalize the first letter of the title?
- 2.Spaces are missing in many places
- 3.Line 112, the name of strain should be in Italic.

Authors' response:

Thank you for the suggestion. We carefully checked to ensure that the above errors have been corrected.

4.3 Reviewer#3 comments:

The results should be re-organized. It is better to have a separate result for antibiotic resistance genes in Line 200-207.

Authors' response:

Thank you for the suggestion. Considering that our research object is the *bla*_{CTX-M-55} gene, the distribution of other β -lactamase coding genes is shown in the Supplementary Table S2. The position in the re-uploaded manuscript is Line 204-211.

4.4 Reviewer#3 comments:

Phylogenetic analysis should contain more strains from China.

Authors' response:

Thank you for the suggestion. According to your suggestion, we downloaded other Chinese *Salmonella* genome data from EnteroBase, and modified the phylogenetic tree content accordingly, which changes in line 219-231, 705-712, and figure 3, and highlighted in yellow.

4.5 Reviewer#3 comments:

Many grammar problems.

Authors' response:

Thank you for the suggestion. The manuscript uploaded again has used the language editing service.

4.6 Reviewer#3 comments:

Figure should be more clear.

Authors' response:

Thank you for the suggestion. According to your suggestion, we have made corresponding changes in Fig. 3.

April 1, 2022

Dr. Xiaoyan Li
Department of Clinical Laboratory, Fifth Affiliated Hospital, Southern Medical University
Guangzhou
China

Re: Spectrum02771-21R1 (**Prevalence of Chromosomally Located *bla*_{CTX-M-55} gene in *Salmonella* Typhimurium ST34 clonal strains from Conghua District of Guangzhou, China**)

Dear Dr. Xiaoyan Li:

The production team has been notified of the new title. Please confirm that it has been changed once you receive your proofs.

New title: Identification of Chromosomally Located *bla*_{CTX-M-55} in *Salmonella* Typhimurium ST34 isolates Recovered From A Tertiary Hospital in Guangzhou, China.

Your manuscript has been accepted, and I am forwarding it to the ASM Journals Department for publication. You will be notified when your proofs are ready to be viewed.

Sincerely,

Sandeep Tamber
Editor, Microbiology Spectrum

Journals Department
Supplemental Tables: Accept
Supplemental Figure: Accept